# High Degree of Polymerization of Chitin Oligosaccharides Produced from Shrimp Shell Waste by Enrichment Microbiota Using Two-Stage Temperature-Controlled Technique of Inducing Enzyme Production and Metagenomic Analysis of Microbiota Succession

**DOI:** 10.3390/md22080346

**Published:** 2024-07-28

**Authors:** Delong Pan, Peiyao Xiao, Fuyi Li, Jinze Liu, Tengfei Zhang, Xiuling Zhou, Yang Zhang

**Affiliations:** School of Life Science, Liaocheng University, Liaocheng 252059, China; 2210150302@stu.lcu.edu.cn (D.P.); m19861908982@163.com (P.X.); 15653129517@163.com (F.L.); 13176629777@163.com (J.L.); lcubiozy@163.com (T.Z.)

**Keywords:** directional domestication, enrichment microbiota, metagenomics, high degree of polymerization, chitin oligosaccharides

## Abstract

The direct enzymatic conversion of untreated waste shrimp and crab shells has been a key problem that plagues the large-scale utilization of chitin biological resources. The microorganisms in soil samples were enriched in two stages with powdered chitin (CP) and shrimp shell powder (SSP) as substrates. The enrichment microbiota XHQ10 with SSP degradation ability was obtained. The activities of chitinase and lytic polysaccharide monooxygenase of XHQ10 were 1.46 and 54.62 U/mL. Metagenomic analysis showed that *Chitinolyticbacter meiyuanensis*, *Chitiniphilus shinanonensis*, and *Chitinimonas koreensis*, with excellent chitin degradation performance, were highly enriched in XHQ10. Chitin oligosaccharides (CHOSs) are produced by XHQ10 through enzyme induction and two-stage temperature control technology, which contains CHOSs with a degree of polymerization (DP) more significant than ten and has excellent antioxidant activity. This work is the first study on the direct enzymatic preparation of CHOSs from SSP using enrichment microbiota, which provides a new path for the large-scale utilization of chitin bioresources.

## 1. Introduction

Chitin polymerized from N-acetylglucosamine (GlcNAc) is widely found in the exoskeletons of crustaceans (e.g., shrimps and crabs), insects, and the cell walls of fungi. Chitin is the most abundant nitrogen-containing natural biopolymer on earth [1]. Although about 10^11^ tons of chitin are produced globally each year, large accumulation in nature is not a problem, mainly due to the degradation of chitin by microorganisms. The formation and degradation of chitin play an important role in the carbon and nitrogen cycle of the earth. With the booming of the global fishing industry, about 6–8 million tons of crab, shrimp, and lobster shell waste are generated every year [2]. These seafood wastes have not been properly treated, which not only causes a waste of resources but also creates challenges related to environmental protection. Chitin and its deacetylated product, chitosan, extracted from shrimp and crab shell residues, are biocompatible, biodegradable, nontoxic, and antibacterial. Chitin and chitosan have been used in agriculture, medicine, food, cosmetics, and the textile industry [3,4]. However, the water insolubility of chitin and chitosan limits their wide application [5]. Chitin oligosaccharides (CHOSs) are a high value-added degradation product of chitin. They have antibacterial, anti-inflammatory, and immune protective activities, intestinal regulation activity, and other biological activities. In particular, their excellent solubility gives them a broad application prospect [6,7,8]. As a positively charged basic amino oligosaccharide, the efficient preparation and application of CHOSs can provide a green method for the high value-added utilization of seafood waste. Currently, CHOSs are obtained through two main methods: chitin degradation and biosynthesis [9,10]. Chitin degradation uses chitin as raw material to prepare CHOSs by physical, chemical, and biological degradation methods. Biodegradation can overcome the disadvantages of physical and chemical methods, such as non-environmental protection and poor quality. It is the trend for CHOS production and preparation in the future [11]. However, CHOSs have not yet been fully industrialized due to the poor activity of the single enzyme, low production efficiency, and the limitation of separation methods, thus hindering its biological activity research and application promotion [12].

So far, much experimental data on the enzymatic preparation of CHOSs has been accumulated [13]. The research results on the structure and mechanism of action of chitin-degrading enzymes significantly increased the possibility of controlling the depolymerization process of chitin. Theoretically, various forms of oligosaccharides can be prepared by selecting different substrates and hydrolases [14]. However, most of the existing preparation theories and production methods of CHOSs are based on chitin and its non-crystalline chitin as substrates, and most of the products obtained are low-molecular-weight CHOSs with a degree of polymerization (DP) below 10 [15]. How to realize the direct enzymatic utilization of crystalline chitin and even natural chitin materials, such as shrimp and crab shells, has become a hot issue in the green and efficient preparation of CHOSs. 

Shrimp and crab shells can be completely degraded by chitin-degrading enzymes produced by microorganisms in the natural environment, such as chitin, chitosanase, and lytic polysaccharide monooxygenase (LPMO), through synergistic action [1]. Under the action of a specific multi-enzyme system, high-efficiency CHOS preparation from shrimp and crab shells can be realized. Currently, the cognition of the synergistic action mode of chitin-degrading enzymes mainly depends on the summary of the enzymatic properties and mechanism of action of a single enzyme. However, this type of study is relatively scarce. It has become a major bottleneck in resolving the synergistic mechanism of chitin-degrading enzymes due to the complexity of the multi-enzymatic degradation process of chitin and the lack of samples. Obtaining microbial communities that degrade biomass efficiently from nature has become a new idea and scheme to solve the problems of the degradation and biotransformation of complex substances [16,17]. Enrichment, domestication, and other technologies are effective methods to obtain microbial communities, which play an important role in the culture of extreme microorganisms and special microbiota [18,19,20]. With the rapid development of high-throughput sequencing technology and modern mass spectrometry (MS) technology, metagenome and metaproteome have been applied to the analysis of microbial community functions and mechanisms. Existing metagenomic studies have shown that although the diversity of microbial communities tends to decline during the enrichment process [21,22], it is conducive to the recovery and enrichment of functional strains, especially dormant cells [23]. Therefore, the combined application of enrichment, domestication and metagenomic technologies has great potential in improving the performance of functional microbiota and analyzing potential mechanisms.

In this study, the soil sample (named LNM) was enriched with chitin powder (CP) and shrimp shell powder (SSP) as substrates by using directional enrichment technology to obtain the acclimated bacteria with the ability to degrade SSP to produce CHOSs with a high DP. With the help of metagenomics technology, the community composition and succession rules of the original samples and the enrichment microbiota were systematically compared. The results provide new insights for the enrichment and domestication in improving the performance of functional microbiota and laid the foundation for studying the synergistic mechanism between chitin-degrading enzymes and the large-scale production of CHOSs.

## 2. Results and Discussions

### 2.1. Domestication of SSP-Degrading Microbiota

The previously established enrichment and acclimation technology was used to conduct a two-stage directional culture of LNM in soil samples [24]. After 20 experimental cycles of enrichment with CP as the substrate, a microbial community (LNM20) with high CP degradation ability was obtained. During the enrichment process, the activities of chitinase and LPMO gradually increased with the increase in enriched experimental cycles (Figure 1a), and the chitinase and LPMO enzyme activities of LNM20 were 0.76 and 34.98 U/mL, respectively. LNM20 was inoculated onto agar plates with CP as the substrate. After culturing, the CP around the colony dissolved and disappeared, forming a visible dissolution circle (Figure 1b), indicating that LNM20 has good CP degradation ability. LNM20 was further acclimated with SSP as the substrate to obtain a high-efficiency degradation bacteria group with SSP degradation performance. After ten experimental acclimation cycles, an enrichment microbiota (XHQ10) that can directly degrade SSP was obtained. The activities of chitinase and LPMO increased during acclimation, but the upward trend gradually slowed. The highest enzyme activities were 1.46 and 54.62 U/mL, which were 1.92 and 1.56 times that of LNM20. Under the condition of liquid fermentation, XHQ10 could degrade most of the powdery SSP in 3 days (Figure 1c,d). Therefore, XHQ10 has a good SSP degradation ability.

### 2.2. Comparison of Chitin Degradation Enzyme Activities of Enrichment Microbiota

Shrimp and crab shells can be completely degraded by microorganisms in the natural environment, and this process is completed by chitin-degrading enzymes produced by microorganisms. The synergistic mechanism of chitin degradation by LPMO and chitinase has opened up a new strategy for the efficient production of CHOSs [25]. The activities of the chitin enzyme and LPMO were determined at different fermentation time points to compare the ability of LNM, LNM20, and XHQ10 to produce the chitin enzyme and LPMO. The results are shown in Figure 2a. The fermentation enzyme production processes of LNM, LNM20, and XHQ10 showed significant differences. The activities of chitinase and LPMO in the whole fermentation process of LNM were close to 0. The chitinase and LPMO activities of LNM20 and XHQ10 were low at the initial stage of fermentation (0–24 h), and then the activity of LPMO showed an obvious upward trend. In contrast, the chitinase activity increased at a greater rate at 60 h. When the fermentation time was 84 h, the two enzyme activities of LNM20 and XHQ10 reached the maximum value, and then the enzyme activities showed a slow downward trend. XHQ10 and LNM20 had the highest chitinase activity of 1.41 and 0.70 U/mL, respectively, and the highest LPMO activity of 55.58 and 34.31 U/mL, respectively. XHQ10 had a higher chitin-degrading enzyme activity than LNM20. The proteins contained in the supernatants of LNM20 and XHQ10 after 84 h of fermentation were analyzed by SDS-PAGE. Figure 2b shows that the proteins secreted by LNM20 and XHQ10 differed greatly in type and concentration, and the number of proteins secreted by XHQ10 that accounted for a higher content was significantly higher. Therefore, enrichment can effectively improve the activities of chitinase and LPMO, and the substrate transition from CP to SSP significantly changes the species and concentration of proteins secreted by the microbiota.

### 2.3. Comparison of Fermentation Products of Enrichment Microbiota

The products obtained by fermentation of LNM20 and XHQ10 after 84 h were analyzed by HPLC, and the results are shown in Figure 3a. With reference to the peak times of DP1-6 in the CHOS standard, the LNM20 and XHQ10 products contained CHOSs at different DPs [26]. The LNM20 product contained DP1-DP6 CHOSs, and the XHQ10 product contained DP3-6 CHOSs. The LNM20 and XHQ10 products still had multiple response peaks after the DP6 standard, and the products were presumed to contain CHOSs with higher DPs. MALDI-TOF MS was used based on HPLC analysis to understand the difference between LNM20 and XHQ10 hydrolysates. As shown in Figure 3b,c, the main peak in the MALDI TOF MS spectrum in positive mode, was attributed to CHOSs with different DPs. The results proved that LNM20 and XHQ10 could degrade SSP effectively and produce CHOSs. The MALDI-TOF MS spectrum of LNM20 showed that CHOSs with a DP of 2–6, determined by HPLC, and CHOSs with a DP of 7–10 were captured. Similarly, CHOSs with DPs of 7–13 were detected in the MALDI-TOF MS spectra of XHQ10. The hydrolysate of XHQ10 contained more CHOSs with high DPs than that of LNM20. 

Chitin degradation by chitinase (endochitinase and exochitinase) is an important way to produce CHOSs [27]. However, the DP of CHOSs obtained by chitin enzymes is mostly below 10 [2]. Studies have shown that the biological activity of CHOSs is closely related to their physical and chemical parameters, such as DP, acetylation degree, and charge distribution [28,29]. CHOSs with slightly higher DPs (DP 5–10) have higher activity in anti-tumor activity, superoxide anion scavenging activity, and immune activation than CHOSs with low DPs (DP 2–4) [13]. High DP CHOSs prepared by XHQ10 are of great significance in promoting CHOS application research.

### 2.4. Community Composition and Succession Analysis during Acclimation

#### 2.4.1. Dataset Overview

Three parallel metagenomic sequences were performed for each of LNM, LNM20, and XHQ10 to reveal the changes in the composition of microbial communities during the enrichment and acclimation of chitin-degrading bacteria. A total of 74,926,897 (22.48 GB) RawReads were obtained for three samples of LNM, with an average of 6.10 GB RawReads per sample; a total of 70,840,368 (21.25 GB) RawReads were obtained for three samples of LNM20, with an average of 7.08 GB RawReads per sample; and XHQ10 received a total of 61,019,577 (18.31 GB) RawReads for three samples, with an average of 7.49 GB RawReads per sample. After low-quality data were removed and host contamination was filtered out, 71,610,263, 67,902,878, and 58,569,366 clean reads were obtained for samples of LNM, LNM20, and XHQ10, respectively, (Table 1).

#### 2.4.2. Community Composition and Taxonomic Changes

The similarity and uniqueness of species composition among sample groups can be intuitively expressed by comparing the number of common and unique species among LNM, LNM20, and XHQ10. As shown in Figure 4a, 68 species were common in the domestication process. In addition, 279 of the 555 species obtained from the 2811 species included in LNM after 20 enrichments were absent from the species composition before domestication. Among the 208 species obtained by XHQ10, 45 species reappeared after the disappearance of LNM20, and 63 species were specific to XHQ10 (Appendix A). The results showed that enrichment and domestication significantly changed the number and species composition of OTUs. The microbial community structure of three sample phylum and genus levels were analyzed. At the phylum level, Pseudomonadota and Actinomycetota were the most abundant taxonomic groups in LNM, accounting for 33.78% and 26.34%, respectively, whereas LNM20 and XHQ10 were dominated by Pseudomonadota, accounting for 98.51% and 99.95%, respectively (Figure 4b,c, Appendix A). At the genus level, *Chitinolyticbacter*, *Laribacter*, *Sinorhizobium*, *Chitiniphilus*, *Streptomyces*, *Herbaspirillum*, *Chitinimonas*, *Microvirga*, and *Stenotrophomonas* were the dominant types in the three samples (Figure 4b,d, Appendix A). Significant differences between the samples at all levels are shown in Figure 4b–d. These results suggest significant changes in the community during enrichment and acclimation.

#### 2.4.3. Excavation of Key Microbiota

The species level and abundance of each macrogenomic sample were analyzed using the linear discriminant analysis effect size method [30] to identify further the microbiota in which the domestication process plays a role. The branching diagram contains six levels from phylum to species, revealing iconic microbiota, with three bacterial taxa enriched in LNM20, six bacterial taxa enriched in XHQ10, and thirty-three bacterial taxa enriched in LNM (Figure 4e). The specifications *Stenotropionas maltohilia*, genus *Stenotropionas*, and family Xanthomonadaceae may play important roles in LNM20, whereas the specifications *Chitolyticbacter meiyuanensis*, genus chitolytic bacteria, family Neisseriaceae, order Neisseriales, class Betaprotecteria, and phylum Pseudomonas may play important roles in XHQ10. A species-level volcano plot (Appendix A) showed that compared with LNM, *Laribacter hongkongensis*, *Chitinolyticbacter meiyuanensis*, *Chitiniphilus shinanonensis*, *Chitinimonas koreensis,* and *Jeongeupia* sp. USM3 were significantly upregulated in LNM20. The top three bacterial groups significantly upregulated in XHQ10 were *Chitinolyticbacter meiyuanensis*, *Chitiniphilus shinanonensis*, and *Chitinimonas koreensis*. *Stenotrophomonas maltophilia* contains nine genes related to chitin turnover [31]. The *Jeongeupia* species has a complete chitin degradation system and shows good chitin degradation performance [32,33]. Although no detailed report on chitin degradation by *Jeongeupia* sp. USM3 is available, 11 potential chitinase genomes were found in its genes [34]. *Laribacter hongkongensis* has chitinase and chitin deacetylase activities, but its chitin degradation ability is relatively weak [35]. *Chitinolyticbacterium meiyuanensis*, *Chitiniphilus shinanonensis*, and *Chitinimonas koreensis* have good chitin degradation ability [36,37,38]. In summary, the proportion of genera related to chitin degradation gradually increased during the enrichment process, with an increase in the enrichment of chitin-degrading bacteria in LNM20 and ultimately, the species in XHQ10 were dominated by chitin-degrading bacteria.

#### 2.4.4. Analysis of Microbiota Diversity

PCoA analysis was performed based on Bray–Curtis distance to determine the diversity and community structure differences of the whole dataset [39]. Figure 4f shows that three parallel datasets from three samples were clustered, with some overlap and dispersion within each cluster. LNM showed considerable dispersion, LNM20 was relatively concentrated, and XHQ10 completely overlapped. The differences in microbial diversity among LNM, LNM20, and XHQ10 samples were revealed. The LNM20 had low abundance and diversity, and the XHQ10 had even lower abundance and diversity.

#### 2.4.5. Functional Annotation Analysis

During acclimation, the microbiota can encode various complex Carbohydrate-Active enZYmes (CAZymes), and these CAZymes contain many enzymes involved in chitin degradation. The relative abundance changes of Glycoside Hydrolase (GHS), GlycosylTransferases (GTs), Polysaccharide Lyases (PLs), Carbohydrate Esterases (CEs), Auxiliary Activities (AAs), and Carbohydrate-Binding Modules (CBMs) in the three samples are summarized in Figure 5a to understand the differences at the gene level in the domestication process. In previous studies, GH, CBM, and AA have been identified as the main CAZymes involved in chitin degradation [40,41]. The GH family is the main component of CAZyme involved in chitin degradation. They can directly hydrolyze the glycosidic bonds of carbohydrate CBMs, which are not directly involved in catalysis, but they help relax the crystal structure of the substrate, improving the overall catalytic efficiency of chitin degradation [40]. The AA family is a newly introduced CAZyme class in recent years, and the multi-enzyme synergism with GHs has an important potential for application in biorefineries. It has become a new direction for developing chitin resources [25]. The relative abundance of GHs was maintained at a high level in the three samples, and no obvious change was found in the relative abundance during domestication (Figure 5a and Appendix A). The relative abundance of CBMs significantly increased at LNM20 and further increased at XHQ10, indicating that the domestication process increased the relative abundance of CBMs (Figure 5a and Appendix A). A notable detail is that the relative abundance of AAs in XHQ10 increased to some extent compared with that of LNM and LNM20 (Figure 5a and Appendix A).

Figure 5b shows the CAZyme heatmap of the top 20 relative abundances. Significant differences were observed in the CAZyme family with the change in the enrichment process. Among these CAZyme families, the GH18 family has chitinase, exo-β-1,4-n-acetylglucosaminidase, and β-n-acetylhexosaminidase activities; the GH19 family has chitinase and exochitinase activities; and the GH20 family has β-1,6-n-acetylglucosaminidase and β-n-acetylhexosaminidase activities. The chitin-degrading enzymes of the three GH families, especially the GH18 family of chitinases, play an important role in degrading chitin [42]. Its relative abundance increased significantly during acclimation (Figure 5b and Appendix A). CBM5 and CBM12 are distantly related, and both have chitin-binding functions that promote the hydrolysis of chitin, whose relative abundance increased significantly during acclimation (Figure 5b and Appendix A). The AA10 family has lytic chitin monooxygenase activity, which plays an important role in synergizing other chitin-degrading enzymes to crystallize chitin [25]. AA10s were significantly enriched in LNM20 and more abundant in XHQ10.

In general, the enrichment of the CAZymes family related to chitin degradation indicates that the hydrolysis ability is improved, which also shows that the domestication enrichment effect is significant. *Chitinolyticbacter meiyuanensis*, *Chitiniphilus shinanonensis*, and *Chitinimonas koreensis* are the main contributors of CAZymes in chitin degradation. Therefore, changes in microbial communities during domestication cause changes in the enzyme family, further improving the chitin degradation ability of enrichment microbiota.

### 2.5. Optimization of Enzyme Production and Degradation Conditions of SSP by XHQ10 and Characterization of Products

#### 2.5.1. Induction of Enzyme Production and Construction of Two-Stage Temperature Control Technology

Except for a few strains, chitinase expression is constitutive and inducible, and most microbial chitinases belong to inducible enzymes. The expression of the microbial chitinase gene significantly increased when it was induced by chitin and its degradation products (such as CHOSs, Nacetylglucosamine, and D-glucosamine) [43,44]. XHQ10 fermentation broth contains chitin and its degradation products, so XHQ10 (XFP) fermentation products were tried as inducers. After XHQ10 was cultured for 12 h, CP, CC, SSP, and XFP (calculated as CHOSs) were added at a final concentration of 1g/L to evaluate their effect on the induction of the enzyme production of XHQ10. By evaluating the induction results, CC and XFP were selected as inducers of XHQ10. The two-stage temperature-controlled fermentation process curve of XHQ10 with XFP and CC as inducers is shown in Figure 6a. The chitinase activity was lower in the early stage of XFP induction than in the control, but the chitinase activity increased significantly after 24 h. The chitinase activity of XFP as an inducer reached a maximum value of 1.61 U/mL at 60 h, which was 1.34 and 2.78 times higher than that of CC as an inducer and control, respectively. After the temperature was adjusted to 50 °C, the chitinase activity of XFP as an inducer showed a slight downward trend. Although the chitinase activity induced by control and CC as inducer still increased, it was consistently lower than that of XFP as an inducer.

#### 2.5.2. Antioxidant Activity of XHQ10 Hydrolysates

CP, CC, and SSP were selected as substrates for enzymatic hydrolysis at 30 °C and 50 °C to investigate the degradation effect of XFP-induced XHQ10 on different substrates at different temperatures. The lyophilized hydrolysates of the three substrates were named CHOS30-CP, CHOS30-CC, and CHOS30-SSP at 30 °C, and the lyophilized hydrolysates at 50 °C were named CHOS50-CP, CHOS50-CC, and CHOS50-SSP. Reactive oxygen species (ROS) are oxygen-active molecules formed during metabolism. Excess ROS and its oxidative stress can cause damage to the body, inducing cancer, cardiovascular disease, and inflammatory diseases [45]. Antioxidants can delay or block the oxidation process to reduce the damage caused by oxidative stress. Existing studies have shown that CHOSs have a certain antioxidant activity [8,46,47]. Therefore, antioxidant capacity is an important indicator of CHOSs. A total antioxidant capacity (T-AOC) assay was performed on lyophilized enzymatic hydrolysates obtained at 30 °C and 50 °C. As shown in Figure 6b, all enzymatic hydrolysates had a certain antioxidant capacity. The order of T-AOC was CHOS50-SSP > CHOS50-CP > CHOS50-CC. The changing trend of T-AOC of the hydrolysates obtained by the enzymatic hydrolysis of the three substrates at 30 °C was the same as that obtained at 50 °C, but the T-AOC of the three hydrolysates was lower than that of CHOS50-CC.

#### 2.5.3. MALDI-TOF/TOF MS Analysis of XHQ10 Enzymatic Hydrolysates

The CHOS50-SSP obtained after 8 h of enzymatic hydrolysis at 50 °C was analyzed by MALDI-TOF/TOF MS and compared with CHOS30-SSP. Figure 6b,c show that CHOS30-SSP and CHOS50-SSP were composed of multiple types of CHOSs. The CHOS30-SSP had a higher percentage of CHOSs, with DPs of 5, 6, 7, and 12, whereas the CHOSs comprising CHOS50-SSP had a higher degree of aggregation, and most of these were CHOSs with DPs > 10. The highest degree of aggregation that could be detected was 28, which is a rarity in studies of enzymatically dissociated gibberellic acid materials. 

XFP was selected as the inducer of XHQ10, and a two-stage process technology was constructed, which induced enzyme production culture at 30 °C and the enzymatic hydrolysis of SSP at 50 °C. This technology can effectively improve the enzyme production level of XHQ10 and effectively degrade SSP to obtain CHOSs with good antioxidant activity and a high DP, which has significant potential in the high-value utilization of discarded shrimp and crab shells.

## 3. Materials and Methods

### 3.1. Chemical Reagents and Enrichment Microbiota

CP was purchased from Shanghai Yuanye Biotechnology Co., Ltd. (Shanghai, China). Shrimp (*Litopenaeus vannamei*) was purchased from local seafood markets. Soil samples were collected in Liaocheng, Shandong, China (36°43′36″ N, 116°01′67″ E). The shells were obtained, thoroughly washed with running water to remove residual shrimp meat and dried in a 60 °C vacuum oven for 24 h. The dried material was crushed in a pulverizer, and then SSP was prepared through a 40-mesh sieve. Colloidal chitin (CC) was prepared using the previous method [48]. Briefly, dissolve 10 g chitin powder in 200 mL pre-cooled concentrated HCl, stir at 4 °C for 2 h, add 2.5 L pre-cooled deionized water, and continue stirring for 10 min. The suspension was centrifuged at 10,000 rpm at 4 °C for 10 min, and the precipitate was repeatedly washed with deionized water to neutral pH and stored at 4 °C for future use. The remaining reagents were analytical grade reagents, and they can be used without additional purification. In accordance with previous research methods [24], 20 experimental cycles of microbiota (LNM20) were enriched and cultured on CP as the initial enrichment microbiota, and LNM20 was acclimated by SSP. The acclimation medium consisted of 2 g/L peptone, 1 g/L glucose, 0.7 g/L K_2_HPO_4_, 0.3 g/L KH_2_PO_4_, 0.5 g/L MgSO_4_, and 4 g/L SSP. After ten experimental cycles of enrichment, the XHQ10 with the ability to degrade SSP was obtained. Soil samples (LNM), LNM20, and XHQ10 were selected for comparison with the fermentation experiment and metagenomic sequencing analysis.

### 3.2. Comparison of Fermentation Performance of LNM, LNM20, and XHQ10

LNM, LNM20, and XHQ10 were individually fermented for 108 h. The fermentation medium consisted of 1 g/L glucose, 2 g/L peptone, 2 g/L yeast extract, 0.7 g/L K_2_HPO_4_, 0.3 g/L KH_2_PO_4_, 0.5 g/L MgSO_4_, and 4 g/L SSP. The culture conditions were as follows: a 250 mL Erlenmeyer flask was filled with 50 mL of fermentation medium, the inoculation volume was 4% (*v*/*v*), the temperature was 30 °C, the pH was 7.0, and the rotating speed was 200 rpm. The fermentation experiment was set up three times in parallel. During the culture process, 1 mL of fermentation broth was sampled every 12 h to determine chitinase and LPMO activities, and the products were analyzed.

### 3.3. Chitin Enzyme and LPMO Activity Analysis

Fixed-point fermentation broth (1 mL) was centrifuged at 4 °C for 10 min at 12,000× *g*, and the supernatant was used for chitin enzyme and LPMO activity assays. Using the method described by Rojas-Avelizapa [49], the chitinase activity was determined by the DNS method under the condition of the chitinase enzymatic hydrolysis of colloidal chitin to release N-acetyl-D-glucosamine (GlcNAc). The absorbance was recorded at 540 nm. The readings were compared with the GlcNAc standard curve. The linear regression equation for the GlcNAc standard curve was Y = 1.2075X − 0.0887 (R^2^ = 0.9994). The unit chitin enzyme activity (U) was defined as the amount of enzyme required to produce 1 mmol of GlcNAc during 1 min at 37 °C. The activity of LPMO was determined in accordance with a previous method [24]. The secreted proteins of LNM20 and XHQ10 after 84 h of fermentation were analyzed by SDS-PAGE with 5% stacked gel and 12.5% resolved gel in Tris-glycine buffer (pH 8.3).

### 3.4. Analysis of Fermentation and Enzymatic Hydrolysis Products

When sampling at a specific time or after reaching the enzyme reaction time, an equal amount of acetonitrile was added to the fermentation broth or enzyme reaction system to terminate the fermentation or enzyme reaction and centrifuged at 4 °C at 12,000× *g* for 10 min through membrane treatment. Samples were separated on a Prevail Carbohydrate ES column (4.6 × 250 mm) at a column temperature of 30 °C, a flow rate of 1.0 mL/min, and a detection wavelength of 195 nm. The mobile phase was acetonitrile and water, and the detection method was as follows: 0 min, acetonitrile: water (75:25, *v*/*v*); 7 min, acetonitrile: water (75:25, *v*/*v*); 8 min, acetonitrile: water (65:35, *v*/*v*); 15 min, acetonitrile: water (65:35, *v*/*v*); 16 min, acetonitrile: water (75:25, *v*/*v*); and 22 min, acetonitrile: water (75:25, *v*/*v*). The detection time was 30 min. 

An appropriate amount of freeze-dried enzymatic hydrolysate was obtained, dissolved in ultrapure water with 2,5-DHB as a substrate, and analyzed using UltrafleXtreme MALDI-TOF/TOF (Bruker Daltonics GmbH, Bremen, Germany) forward collection mode. The collection adopted the reflective type, the acceleration voltage was 20 kV, the reflective layer voltage was 21.1 kV, and the collection range was *m*/*z* 400–6000.

### 3.5. Metagenomic Sequencing

The total DNA of LNM, LNM20, and XHQ10 was extracted using the E.Z.N.A. Soil DNA Kit (Omega Bio-Tek, Norcross, GA, USA). The metagenomic library was prepared and sequenced by Shenzhen Microecology Technology Co., Ltd. (Shenzhen, China) using the HiSeq2000 platform (Illumina PE150, San Diego, CA, USA).

### 3.6. Bioinformatics and Data Analysis

Kraken2 was used to compare with the self-built microbial nucleic acid database (screening the sequences belonging to bacteria, fungi, archaea, and viruses in the NCBI NT nucleic acid database and RefSeq whole genome database) to calculate the sequence number of species contained in the sample. Bracken was then used to estimate the actual abundance of species in the sample. Read-based metagenomic species annotation is more comprehensive and accurate than assembly-based species annotation. Starting from quality control and removing the reads of host genes, the reads of each sample were compared to the database (UniRef90) by using HUMAnN 3 3.6 software (based on DIAMOND, and the annotation information and relative abundance table of the CAZY database were obtained in accordance with the corresponding relationship between UniRef90 ID and the CAZY database.

### 3.7. Two-Stage Temperature-Controlled Degradation of SSP by XHQ10 to Prepare CHOSs

XHQ10 was cultured in a culture medium without SSP at 30 °C and 200 rpm. After 12 h, CC and XFP were added and continued to be induced to culture for 60 h. After the temperature was increased to 50 °C, SSP with a final concentration of 8 g/L was added. After eight hours of incubation, it was centrifuged at 4 °C at 12,000× *g* for 10 min. The supernatant was lyophilized, and the CHOS product was obtained.

### 3.8. Analysis of the Antioxidant Activity of CHOSs

The modified FRAP method [50] was used to detect the total antioxidant capacity (T-AOC) of different CHOS samples. The specific operation and analysis steps were carried out in accordance with the specification of the T-AOC Assay Kit (BIOBOX, AKAO012C, Beijing, China).

## 4. Conclusions

In this study, LNM from soil samples was enrichment cultured with CP as a substrate to obtain chitin-degrading microbiota LNM20, and LNM20 was acclimated with SSP to obtain the enrichment microbiota XHQ10 with high-efficiency SSP degradation. The community composition and succession rules of the original samples and enrichment microbiota were systematically compared using metagenomics technology. The high antioxidant activity of CHOSs was obtained by optimizing the conditions of XHQ10 enzymatic hydrolysis and constructing the enzyme induction and two-stage temperature control technology. The results provide new insights for enrichment and acclimation in improving the performance of functional microbiota. They lay the foundation for studying the synergistic mechanism between degrading enzymes in chitin and the large-scale production of CHOSs.

## Figures and Tables

**Figure 1 marinedrugs-22-00346-f001:**
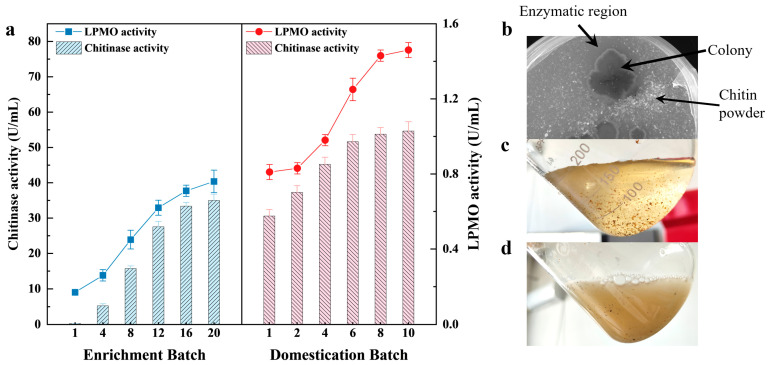
Enrichment effect and degradation performance of different experimental cycles of enrichment microbiota. (**a**) Chitin-degrading enzyme activity of enrichment microbiota in different experimental cycles. (**b**) Enzymatic region degrading CP and producing clear transparent circles on agar plates. (**c**,**d**) State of SSP in the fermentation broth before inoculation with XHQ10 and three days after fermentation.

**Figure 2 marinedrugs-22-00346-f002:**
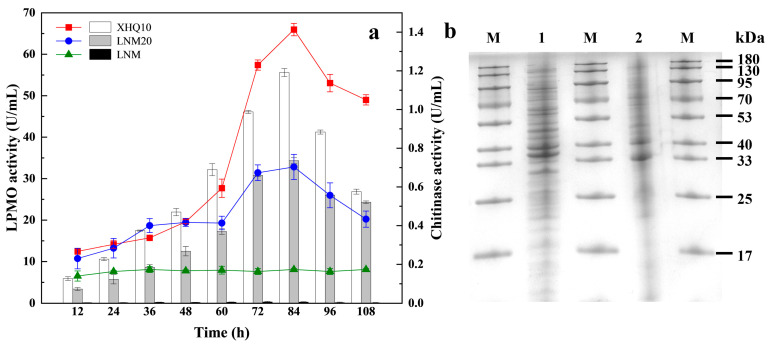
Enzyme production of different bacterial groups during domestication. (**a**) Changes in enzyme activity during different microbial fermentation processes. (**b**) SDS-PAGE analysis of LNM20 and XHQ10 fermentation supernatants. M, molecular mass markers; lane 1, protein contained in XHQ10 supernatant after 84h of fermentation; and lane 2, protein contained in LNM20 supernatant after 84h of fermentation.

**Figure 3 marinedrugs-22-00346-f003:**
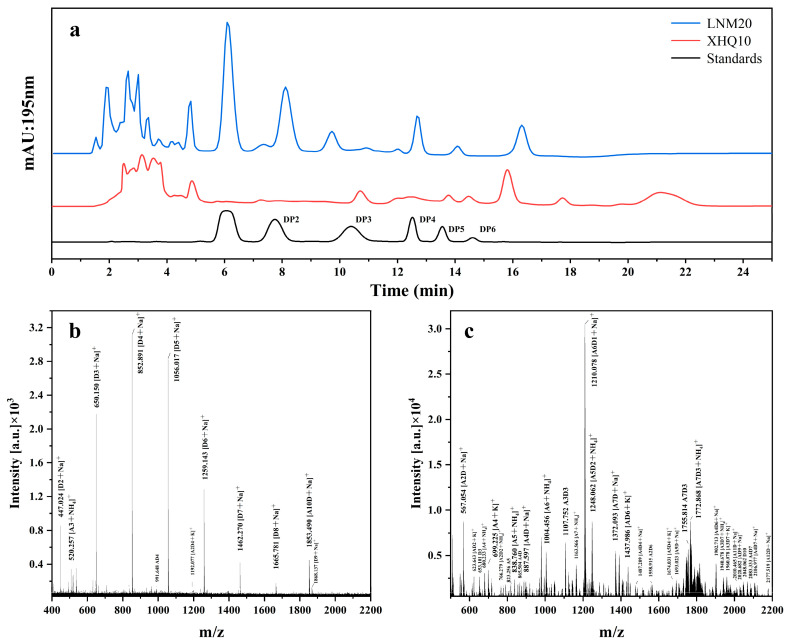
Hydrolysis of CP-, SSP-, and CHOS-generating capacity of LNM20 and XHQ10. (**a**) HPLC profiles of fermentation products of LNM20 and XHQ10. MALDI-TOF MS analysis of LNM20 (**b**) and XHQ10 (**c**) fermentation products.

**Figure 4 marinedrugs-22-00346-f004:**
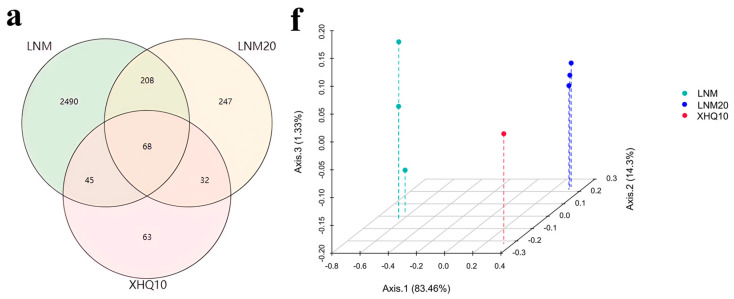
Metagenomic analysis of the microbiota during domestication. (**a**) Venn diagram of OTUs for three samples. (**b**) Three sample species-level phylogenetic trees. The color of the heatmap represents the number of strains contained by each species in the three samples. (**c**) Relative abundance at the phylum level. (**d**) Relative abundance at the genus level. (**e**) Dynamic branching diagram of microbial phylogeny of three samples. Nodes with different colors represent microorganisms that play a key role in the grouping indicated by the color, and yellow nodes indicate that they are not significant. Microbial markers with LDA score ≥ 4 in the three samples are listed. Volcano plot of the differences between LNM and LNM20. (**f**) Between-group analysis of PCoA based on the Bray-Curtis distance. Different colors represent different groups.

**Figure 5 marinedrugs-22-00346-f005:**
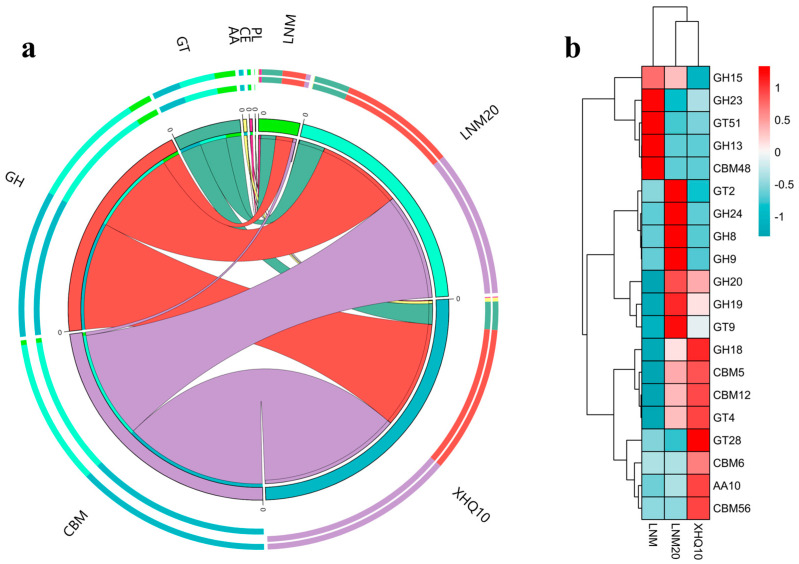
Analysis of CAZyme composition. (**a**) Dynamic changes in CAZyme class-level composition of three samples. (**b**) Heatmap of the top 20 CAZymes’ relative abundance at the family-level of three samples.

**Figure 6 marinedrugs-22-00346-f006:**
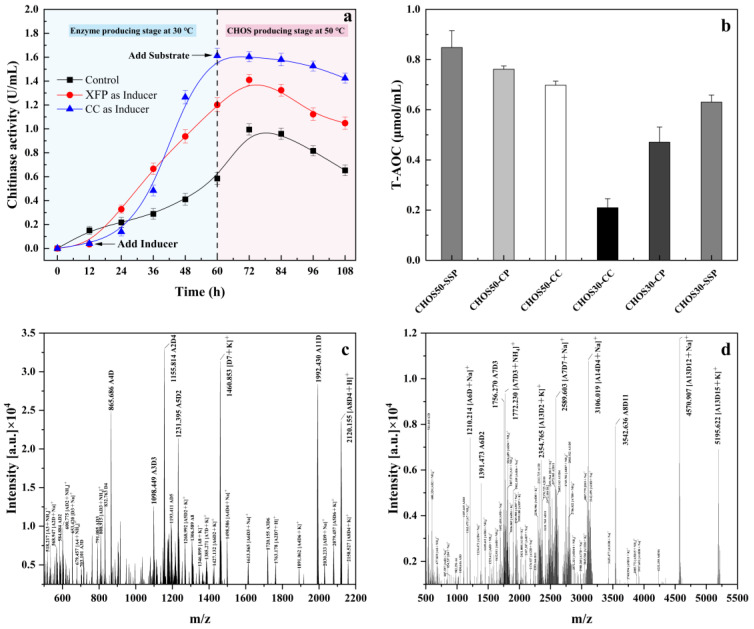
Changes in chitinase enzyme activity under enzyme-producing induction and two-stage temperature control techniques. (**a**) Changes in enzyme activity of different substrates under a two-step mechanism. (**b**) Antioxidant capacity. (**c**) CHOS30-SSP MALDI-TOF MS analysis. (**d**) CHOS50-SSP MALDI-TOF MS analysis.

**Table 1 marinedrugs-22-00346-t001:** Overview of the quality of metagenomic sequencing data.

Sample ID	Grouping	RawReads	Raw Base (GB)	Clean Reads	Cleaned (%)
LNM	SS11	19784882	5.94	18925514	95.66
	SS12	26947750	8.08	25736535	95.51
	SS13	28194265	8.46	26948214	95.58
LNM20	FS21	22424308	6.73	21512100	95.93
	FS22	26312463	7.89	25221987	95.86
	FS23	22103597	6.63	21168791	95.77
XHQ10	FS11	20613776	6.18	19727621	95.7
	FS12	20716845	6.22	19935093	96.23
	FS13	19688956	5.91	18906652	96.03

## Data Availability

The data from the present study are available in the article; further inquiries can be directed to the corresponding author.

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
