# Peer review of "High Degree of Polymerization of Chitin Oligosaccharides Produced from Shrimp Shell Waste by Enrichment Microbiota Using Two-Stage Temperature-Controlled Technique of Inducing Enzyme Production and Metagenomic Analysis of Microbiota Succession"

_marinedrugs, 2024, doi:10.3390/md22080346_

Round 1
Reviewer 1 Report
Comments and Suggestions for Authors
This article introduced a very interesting work, just as author mentioned, “The results provide new insights for enrichment and acclimation in improving the performance of functional flora. They lay the foundation for the study of the synergistic mechanism between degrading enzymes in chitin and the large-scale production of COS.”. However, there are too many mistakes in the manuscript, which I can not tolerate. Rather than a final draft, I feel like this manuscript is a first draft that hasn’t been properly examined. In view of these issues, I do not recommend accepting the manuscript.
Including but not limited to the following issues.
1. First of all, the author did not distinguish the chitinooligosaccharides and chitooligosaccharides. Commonly, we use COS stands for chitooligosaccharides, and it is a positively charged basic amino oligosaccharide. In this article, the author used it to stand for chitinooligosaccharides, however, it was not in accordance with the reference cited ([8], [14], [42], [43]...).
2. For the abbreviation, what is XFP? LNM (lines 87 and 360), T-AOC (lines 323 and 424).
3. The result of Figure 1d was 3 or 4 days (lines 113 and 119)?
4. References are missing in many places, for example, lines 98-99, 150-151, 229-230, 280-282.
5. The section of “results and discussion” are lack of discussion. The discussion about reference 26 are far-fetched.
6. What each symbol in Figure 2 represents is not clearly explained.
7. In Figure 3a and line 165, “LNM10” should be “LNM20”.
8. The figures 3b, 3c, 4, 6c and 6d are not clear enough.
9. The content of the article is inconsistent with the results of the Figure 6 (lines 305-310).
10. Some mistakes:
w (line 224), : (line 227), the (line 251), missed . (lines 256 and 277), CNM (line 277), cazyme (line 287), COS50-SSP (line 326), the sentences in lines 310-311 and 334-335, PD (line 339)...
Comments on the Quality of English Language
This article introduced a very interesting work, just as author mentioned, “The results provide new insights for enrichment and acclimation in improving the performance of functional flora. They lay the foundation for the study of the synergistic mechanism between degrading enzymes in chitin and the large-scale production of COS.”. However, there are too many mistakes in the manuscript, which I can not tolerate. Rather than a final draft, I feel like this manuscript is a first draft that hasn’t been properly examined. In view of these issues, I do not recommend accepting the manuscript.
Including but not limited to the following issues.
1. First of all, the author did not distinguish the chitinooligosaccharides and chitooligosaccharides. Commonly, we use COS stands for chitooligosaccharides, and it is a positively charged basic amino oligosaccharide. In this article, the author used it to stand for chitinooligosaccharides, however, it was not in accordance with the reference cited ([8], [14], [42], [43]...).
2. For the abbreviation, what is XFP? LNM (lines 87 and 360), T-AOC (lines 323 and 424).
3. The result of Figure 1d was 3 or 4 days (lines 113 and 119)?
4. References are missing in many places, for example, lines 98-99, 150-151, 229-230, 280-282.
5. The section of “results and discussion” are lack of discussion. The discussion about reference 26 are far-fetched.
6. What each symbol in Figure 2 represents is not clearly explained.
7. In Figure 3a and line 165, “LNM10” should be “LNM20”.
8. The figures 3b, 3c, 4, 6c and 6d are not clear enough.
9. The content of the article is inconsistent with the results of the Figure 6 (lines 305-310).
10. Some mistakes:
w (line 224), : (line 227), the (line 251), missed . (lines 256 and 277), CNM (line 277), cazyme (line 287), COS50-SSP (line 326), the sentences in lines 310-311 and 334-335, PD (line 339)...
Author Response
Dear Reviewer:
Thank you for your time and effort in reviewing our manuscript. We were grateful for your thoughtful feedback and valuable insights. We greatly appreciate your positive comments. Those comments are all valuable and very helpful for revising and the important guiding significance of our research. We have studied your comments carefully and have made a correction which we hope meet with approval. We looking forward to seeing the results of your review. Special thanks for your very constructive comments. Our point-by-point replies to the comments are also summarized, and the list is as follows.
1. First of all, the author did not distinguish the chitinooligosaccharides and chitooligosaccharides. Commonly, we use COS stands for chitooligosaccharides, and it is a positively charged basic amino oligosaccharide. In this article, the author used it to stand for chitinooligosaccharides, however, it was not in accordance with the reference cited ([8], [14], [42], [43]...)
Reply and revision: Thank you for your kind suggestion. We very much agree with your point of view. We apologize for our lack of rigour in our professional knowledge. We have consulted the literature related to chitinooligosaccharides and chitooligosaccharides based on your suggestion. Chitinooligosaccharides are oligosaccharides composed of β-1,4-linked Nacetylglucosamine and a small amount of D-glucosamine, chitosanoligosaccharides are homo- or hetero-oligomers of Nacetylglucosamine and D-glucosamine (Systems Microbiology and Biomanufacturing, 2023, 3:49–74; Carbohydrate Polymers, 2024, 324:121546). On the basis of following your suggestion and according to the characterization results of the product we obtained, we revised chitooligosaccharides to chitin oligosaccharides and COS to CHOS.
2. For the abbreviation, what is XFP? LNM (lines 87 and 360), T-AOC (lines 323 and 424).
Reply and revision: We greatly appreciate your professional comments on our articles, and depending on the issues you have pointed out, there are several acronyms in our manuscript that may not be articulated at first use. We explain the meaning of abbreviation in the manuscript to make the text more accessible to the reader. (Page 9, lines 314-315; Page 2, line 89; Page 9, line 338)
3. The result of Figure 1d was 3 or 4 days (lines 113 and 119)?
Reply and revision: We sincerely thank you for your question. Figure 1d shows that XHQ10 can degrade most of the powdery SSP within 3 days under liquid fermentation conditions. In the main text of the first edition of the manuscript, due to our carelessness, we have changed "4 days" to "3 days". Once again, we apologize for our carelessness. (Page 3, line 115)
4. References are missing in many places, for example, lines 98-99, 150-151, 229-230, 280-282.
Reply and revision: We sincerely thank you for your valuable opinions, and supplementing relevant literature can make our manuscript more convincing. As you suggested, we cite more relevant literature in the article to support it.
5. The section of “results and discussion” are lack of discussion. The discussion about reference 26 are far-fetched.
Reply and revision: Thank you for your kind suggestion. We very much agreed with your point of view. We have removed relevant content and citations from reference 26 in the manuscript.
6. What each symbol in Figure 2 represents is not clearly explained.
Reply and revision: We sincerely appreciate your valuable feedback. In response to your question, we have explained the protein gel electrophoresis diagram in Figure 2, which is added at the end of the figure caption in Figure 2. (Page 4, lines 147, 150-152)
7. In Figure 3a and line 165, “LNM10” should be “LNM20”.
Reply and revision: We sincerely thank you for your valuable feedback and professional evaluation of our article. We are deeply sorry that Figure 3 was incorrectly annotated due to our mistake. We have corrected this error in the new manuscript by changing "LNM10" to "LNM20". (Page 5, line 176)
8. The figures 3b, 3c, 4, 6c and 6d are not clear enough.
Reply and revision: I apologize for the inconvenience. We reproduced figures 3b, 3c, 4, 6c and 6d, and uploaded these figures in tif format files to the manuscript system. Please download them for review.
9. The content of the article is inconsistent with the results of the Figure 6 (lines 305-310).
Reply and revision: Thank you for your important questions and professional advice about our work. We have rewritten these sentences to avoid ambiguity. (Page 9, lines 320-326)
10. Some mistakes:
w (line 224), : (line 227), the (line 251), missed . (lines 256 and 277), CNM (line 277), cazyme (line 287), COS50-SSP (line 326), the sentences in lines 310-311 and 334-335, PD (line 339)...
Reply and revision: We greatly appreciate your valuable feedback in making a comprehensive revision of our manuscript based on the issues you have pointed out.

Reviewer 2 Report
Comments and Suggestions for Authors
The authors collected soil samples from the environment to screen for chitin degrading microorganisms. In the course of different cultivation cycles with chitin powder as additional carbon source the degrading ability of the culture is enhanced. The enriched samples are analyzed by Metagenomic and -proteomic to identify the composition of chitin degrading enzyme families. Finally, the enriched culture was applied to depolymerize shrimp shell powder into chito-oligosaccharides of optimal length for increased antioxidative activity. Here a two stage temperature fermentation process was developed.
Before I write down my comments according to the line numbering of the manuscript, I would like to discuss your expression “domesticated flora”. I don’t see how the expression domesticated which is normally applied to animals can be applied in the context of enriching a microorganism community to cope with external conditions, like here the provision of a kind of chitin as carbon and nitrogen source. Furthermore, in my point of view there is no significant difference between the enrichment cycle 1 which is called “enrichment” leading to LNM20 and the cultivation batches with SSP instead of CP leading to XHQ10 and called “domestication”. Scientifically sound would be to call both an enrichment and adaption depending on the applied kind of chitin. Additionally, I am surprised by your usage of “flora”. According to the Cambridge English Dictionary flora means either “all the plants of a particular place or from a particular time in history” or “all the bacteria and other organisms that live inside an animal”. I don’t think that any definition is describing the situation in your manuscript. Please reconsider the usage of “domesticated flora” in the context of your article.
L.18: Maybe you can add to the number 10 monomers or monomeric units, to make it clearer.
l.34: I would put “waste” behind the word “shells”
l.39: Please elaborate on the application of Chitin/chitosan you want to mention by “environment”. To me the expression is too general.
l.43: “to” is missing behind “makes it”
l.49: You mention that physical and chemical methods have the drawback of high costs. Please show me by a workflow example that the mentioned methods are causing high-costs.
l.58: hydrolases instead of hydrolase
l.82: metagenomic instead of metagenomics
l.87: The used abbreviation “LNM” is quite special, can you tell me the whole expression it stands for? Second, I am missing in the whole manuscript the location you derived the soil samples from. Please add this at an appropriate position.
l.149: I think replacing “for” by “after 84h” makes the experiment clearer.
l.152 and fig. 3A: Please correct “LNM10” to “LNM20”.
l.191: What means “OTUs”?
l.224: Please reconsider this sentence. Something is missing.
l.227: Please erase “:”
Fig. 4: A) It is hardly possible to see in the figure, what should be exemplified by it. It must be significantly bigger. Less graphs should occupy the page width. B) Please erase the second closing bracket in l.238.
L.250: Please introduce to the reader the abbreviation “CAZyme”.
l.251: Please start a new sentence with a capital letter.
l.252: Please enhance the readability by mentioning the whole names of the enzyme families “GHS, GTS,…” (or you can refer to the last column of Table S3.)
l.277: I expect “CNM12” should be “CBM12”.
Fig. 5: In l.292 of the legend it is mentioned that the figure shows the relative abundance of CAZymes (please correct this in l.291) at different stages. What do you mean by different stages? It can be easily mixed up with the growth phase of the colonies, which should be roughly the same.
l.300: Only the word “inducer” is too short. Although you mention the inducers in the next sentence, I think mentioning them before as examples would create a better connection between both sentences. Additionally, it would help the reader to follow the experimental flow.
l.300ff: Please reconsider this sentence. It seems incomplete.
l.305: chitin”ase”
l.326: A)“SSP” has to be replaced by “CC” according to fig. 6b. B) The statement of the next sentence is not fitting to graph 6b. Please either revise the sentence or explain how the statement should be understood in the context of fig. 6b.
chapter 3.1 and 3.2: It is hard for the reader to understand in details the experimental workflow. Although you refer to literature 45 (where it is explained with enough details), it would be good to give enough details to follow without the necessity of lit. 45.
For example, it should become more clear, that a batch can be replaced by the synonym “experimental cycle”.
Please revise in both chapters the chemical formulas.
What do you mean by “filing volume” in l.367? Volume of the vessel (By the way what kind of vessel?) or reaction/culture volume?
What do you mean by “small amount” in l.369? What was the exact volume you withdraw during the whole cultivation? How many fixed time points (l.370) per cultivation experiment. Only with the exact values the reader can derive how important the volume reduction by sample taking was.
l.380: I suggest to replace “for 1 min” by “in” or “during 1 min”
l.386: Can you state more precisely after what reaction time you stopped the reaction by addition of acetonitrile?
l.400: “were” instead of “was”
Additional gel picture: Can you provide a legend explaining what is shown in the lanes marked by an X. Please explain, why did you upload this gel beside fig. 2b?
After tackling the open questions and points raised, I would recommend the manuscript for publication.
Comments on the Quality of English Language
There are some minor points raised concerning the English spelling. Please refer to the complete list of comments/suggestions to improve the article.
Author Response
Dear Reviewer:
Thank you for your time and patience. Thank you for your detailed review of the manuscript. We are greatly appreciated it. Thank you very much for your professional comments and suggestions. Those comments are all valuable and helpful for revising and improving our paper. We have tried our best to revise and improve the manuscript and made some changes in the manuscript according to your comments and hope that the corrections will meet with approval. Our point-by-point replies to the comments are also summarized, and the list is as follows.
1. Before I write down my comments according to the line numbering of the manuscript, I would like to discuss your expression “domesticated flora”. I don’t see how the expression domesticated which is normally applied to animals can be applied in the context of enriching a microorganism community to cope with external conditions, like here the provision of a kind of chitin as carbon and nitrogen source. Furthermore, in my point of view there is no significant difference between the enrichment cycle 1 which is called “enrichment” leading to LNM20 and the cultivation batches with SSP instead of CP leading to XHQ10 and called “domestication”. Scientifically sound would be to call both an enrichment and adaption depending on the applied kind of chitin. Additionally, I am surprised by your usage of “flora”. According to the Cambridge English Dictionary flora means either “all the plants of a particular place or from a particular time in history” or “all the bacteria and other organisms that live inside an animal”. I don’t think that any definition is describing the situation in your manuscript. Please reconsider the usage of “domesticated flora” in the context of your article.
Reply and revision: Thank you for your detailed and thoughtful review of the manuscript, we are greatly appreciated. Your insight and expertise have been invaluable in helping me to understand the strengths and weaknesses of our work. Follow your suggestion, we have tried our best to retrieve relevant literature and analyze it for comparison. We strongly agree with your views on "domestication" and "flora". Our main research content is to obtain microbiota with excellent shrimp shell powder degradation ability by directional enrichment technology. We very much agree with you that the word “domestication” and “flora” is an inappropriate term in manuscripts. We have corrected all “domination” and “flora” to “enrichment” and “microbiota”, the corresponding expression has also been revised.
2. L.18: Maybe you can add to the number 10 monomers or monomeric units, to make it clearer.
Reply and revision: Thank you for your kind suggestion. We have revised it. (Page 1, lines 18-21)
3. l.34: I would put “waste” behind the word “shells”
Reply and revision: Thank you for your kind suggestion. We have revised it. (Page 1, line 35)
4. l.39: Please elaborate on the application of Chitin/chitosan you want to mention by “environment”. To me the expression is too general.
Reply and revision: Thank you for your valuable comments. We apologize for our lack of rigour in our professional knowledge. We have deleted “environment”. (Page 1, lines 39-40)
5. l.43: “to” is missing behind “makes it”
Reply and revision: Thank you for your kind suggestion. We have revised it. (Page 2, line 44)
6. l.49: You mention that physical and chemical methods have the drawback of high costs. Please show me by a workflow example that the mentioned methods are causing high-costs.
Reply and revision: Thank you for your kind suggestion. Physical and chemical methods have cost advantages in producing CHOS. We are very sorry that our carelessness has caused this mistake. we have corrected the mistake and revised it. (Page 2, lines 49-51)
7. l.58: hydrolases instead of hydrolase
Reply and revision: Thank you for your kind suggestion. We have revised it. (Page 2, line 59)
8. l.82: metagenomic instead of metagenomics
Reply and revision: Thank you for your kind suggestion. We have revised it. (Page 2, line 83)
9. l.87: The used abbreviation “LNM” is quite special, can you tell me the whole expression it stands for? Second, I am missing in the whole manuscript the location you derived the soil samples from. Please add this at an appropriate position.
Reply and revision: We have previously reported an article "Zhang Y, Pan D, Xiao P, Xu Q, Geng F, Zhang X, Zhou X and Xu H (2023) A novel lytic polysaccharide monoxygenase from enrichment microbiota and its application for shrimp shell powder biodegradation, Front. Microbiol. 14: 1097492.", and the soil samples used in this article continue to be used in the current study, so the name of this soil continues. We have added the "3.1" section of "Materials and Methods" for your proposed location of obtaining soil samples,Soil samples were collected in Liaocheng, Shandong, China (116°0167′N, 36°4,336′E). (Page 10, lines 367-368)
10. l.149: I think replacing “for” by “after 84h” makes the experiment clearer.
Reply and revision: Thank you for your kind suggestion. We have revised it. (Page 4, line 154)
11. l.152 and fig. 3A: Please correct “LNM10” to “LNM20”.
Reply and revision: Thank you for your kind suggestion. We have revised it. (Page 5, line 176)
12. l.191: What means “OTUs”?
Reply and revision: We sincerely thank you for your question. OUT is the abbreviation of "Operational Taxonomic Unit". Each OUT represents a biological species or subspecies. Here, its number can reflect the diversity and abundance of microbial communities, and can clarify the composition of microbial communities and their role in the ecological environment.
13. l.224: Please reconsider this sentence. Something is missing.
Reply and revision: Thank you for your professional advice. We have made corrections in this sentence. (Page 6, lines 234-238)
14. l.227: Please erase “:”
Reply and revision: Thank you for your valuable comments. We have deleted it. (Page 7, line 238)
15. Fig. 4: A) It is hardly possible to see in the figure, what should be exemplified by it. It must be significantly bigger. Less graphs should occupy the page width. B) Please erase the second closing bracket in l.238.
Reply and revision: We apologize for the inconvenience. We recreated the figures and uploaded the tif format files of these figures to the manuscript system. Please download them for review. We removed the second closing bracket. (Page 7, line 249)
16. L.250: Please introduce to the reader the abbreviation “CAZyme”.
Reply and revision: Thank you for your professional advice. We have revised it. (Page 7, lines 261-262)
17. l.251: Please start a new sentence with a capital letter.
Reply and revision: Thank you for your kind suggestion. We have revised it. (Page 7, line 263)
18. l.252: Please enhance the readability by mentioning the whole names of the enzyme families “GHS, GTS,…” (or you can refer to the last column of Table S3.)
Reply and revision: Thank you for your professional advice. We have added detailed information to the manuscript. (Page 7, lines 263-265)
19. l.277: I expect “CNM12” should be “CBM12”.
Reply and revision: Thank you for your kind suggestion. We have revised it. (Page 8, line 290)
20. Fig. 5: In l.292 of the legend it is mentioned that the figure shows the relative abundance of CAZymes (please correct this in l.291) at different stages. What do you mean by different stages? It can be easily mixed up with the growth phase of the colonies, which should be roughly the same.
Reply and revision: We sincerely thank you for your professional evaluation of our manuscript. Different stages are intended to represent three different domestication stages of soil samples, LNM20 and XHQ10, and the expressions in the first edition manuscript are prone to ambiguity. In response to your question, we have made a correction to correct "domesticated flora at different stages" to "three samples". (Page 9, lines 304, 305)
21. l.300: Only the word “inducer” is too short. Although you mention the inducers in the next sentence, I think mentioning them before as examples would create a better connection between both sentences. Additionally, it would help the reader to follow the experimental flow.
Reply and revision: Thank you very much for your professional comments and suggestions. We have revised it. (Page 9, lines 313-317)
22. l.300ff: Please reconsider this sentence. It seems incomplete.
Reply and revision: Thank you very much for your professional comments and suggestions. We have revised it. (Page 9, lines 313-317)
23. l.305: chitin”ase”
Reply and revision: Thank you for your kind suggestion. We have revised it. (Page 9, line 320)
24. l.326: A)“SSP” has to be replaced by “CC” according to fig. 6b. B) The statement of the next sentence is not fitting to graph 6b. Please either revise the sentence or explain how the statement should be understood in the context of fig. 6b.
Reply and revision: Thank you very much for your professional comments and suggestions. We apologize for our imprecision. We have rewritten this section to make the semantics more accurate. (Page 9, line 341-344)
25. chapter 3.1 and 3.2: It is hard for the reader to understand in details the experimental workflow. Although you refer to literature 45 (where it is explained with enough details), it would be good to give enough details to follow without the necessity of lit. 45.
Reply and revision: Thank you for your kind suggestion. We have added details of the experimental workflow to the manuscript. (Page 10, lines 372-376)
26. For example, it should become more clear, that a batch can be replaced by the synonym “experimental cycle”.
Reply and revision: Thank you for your professional advice. We have made corrections in the whole manuscript.
27. Please revise in both chapters the chemical formulas.
Reply and revision: Thank you very much for your professional comments and suggestions. We have revised it. (Page 11, lines 380, 381, 387 and 388)
28. What do you mean by “filing volume” in l.367? Volume of the vessel (By the way what kind of vessel?) or reaction/culture volume?
Reply and revision: Thank you for your professional advice. We have revised it.
A 250 mL Erlenmeyer flask was filled with 50 mL of fermentation medium. (Page 11, lines 388-389)
29. What do you mean by “small amount” in l.369? What was the exact volume you withdraw during the whole cultivation? How many fixed time points (l.370) per cultivation experiment. Only with the exact values the reader can derive how important the volume reduction by sample taking was.
Reply and revision: Thank you for your professional advice. We have revised it. During the culture process, 1 mL of fermentation broth was sampled every 12 h to determine chitinase and LPMO activities, and the products were analyzed. (Page 11, lines 391-393)
30. l.380: I suggest to replace “for 1 min” by “in” or “during 1 min”
Reply and revision: Thank you for your kind suggestion. We have corrected it. (Page 11, line 403)
31. l.386: Can you state more precisely after what reaction time you stopped the reaction by addition of acetonitrile?
Reply and revision: Thank you for your professional advice. We have revised it. (Page 11, lines 408-410)
32. l.400: “were” instead of “was”
Reply and revision: Thank you for your kind suggestion. We have revised it. (Page 11, line 424)
33. Additional gel picture: Can you provide a legend explaining what is shown in the lanes marked by an X. Please explain, why did you upload this gel beside fig. 2b?
Reply and revision: Thank you for your question. We've added legend to gel picture. Fig. 2b is an SDS-PAGE analysis of proteins secreted by LNM20 and XHQ10. The main objective was to examine whether there were differences in species and concentration of proteins secreted by the two enrichment microbiota species. To explain why, XHQ10 has higher chitinase and LPMO activities than LNM20. For your question, we have the following comments: the first X lane is molecular mass markers; The second X lane is protein contained in XHQ10 superatant after 24h of induction; The third X lane is the repeated addition of Lane 2. Three more lanes were added because the gel hole was free at that time, but our manuscript is mainly to compare the enzyme production of the best enzyme production time of the two samples, so only a few lanes in Fig. 2B are presented.

Round 2
Reviewer 1 Report
Comments and Suggestions for Authors
The authors have revised the manuscript satisfyingly. I have no further comments.
Author Response
Dear Reviewer:
Thank you for your valuable comments to improve the readability and scientificity of the revised manuscript.
Reviewer 2 Report
Comments and Suggestions for Authors
I see a significant improvement of the manuscript.
However, fig. 4 is still too small. If I zoom into the graph, I still cannot read the legend of phylum or genus in fig. 4c and 4d, respectively. In fig. 4a you can guess the numbers/values instead of reading. Even worse is the readability of part b, e, f and g of fig. 4.
In my point of view a figure should be self explaining. Furthermore, the parts of fig. 4 show much more of information as can be described in the text of the corresponding paragraphs. It is easily possible that a reader will look with a different perspective like the authors at the figures and can maybe extract from readable figures an aspect which is irrelevant to the authors.
Hence, I strongly encourage the authors again to substantially improve fig. 4, as it is at the heart of the manuscript. Too many information which can be derived from the data analysis perish because of the poor quality/size of the graphs in fig. 4.
Comments on the Quality of English Language
When I read the revised manuscript, I still saw some spelling errors. Another read through the manuscript will reveal most of them.
Author Response
Dear Reviewer:
Thank you for your time and patience. Thank you for your detailed review of the manuscript. We are greatly appreciated it. Thank you very much for your professional comments and suggestions. Those comments are all valuable and helpful for revising and improving our paper. We have tried our best to revise and improve the manuscript and made some changes in the manuscript according to your comments and hope that the corrections will meet with approval. Our point-by-point replies to the comments are also summarized, and the list is as follows.
Fig. 4 is still too small. If I zoom into the graph, I still cannot read the legend of phylum or genus in fig. 4c and 4d, respectively. In fig. 4a you can guess the numbers/values instead of reading. Even worse is the readability of part b, e, f and g of fig. 4.
In my point of view a figure should be self explaining. Furthermore, the parts of fig. 4 show much more of information as can be described in the text of the corresponding paragraphs. It is easily possible that a reader will look with a different perspective like the authors at the figures and can maybe extract from readable figures an aspect which is irrelevant to the authors.
Hence, I strongly encourage the authors again to substantially improve fig. 4, as it is at the heart of the manuscript. Too many information which can be derived from the data analysis perish because of the poor quality/size of the graphs in fig. 4.
Reply and revision: Thank you for your thoughtful review and constructive feedback provided. You propose to improve the clarity of Figure 4. We agree with your proposal and incorporate the suggested modifications into the manuscript. We revised the figure to enhance readability, and we appreciate the valuable suggestion. (Pages 7 and 8, lines 244-249)
When I read the revised manuscript, I still saw some spelling errors. Another read through the manuscript will reveal most of them.
Reply and revision: Thanks for your suggestion. We feel sorry for our poorwritings. However, we do invite a friend of us whois a native English speaker from USA help polish ourarticle. And we hope the revised manuscript could beacceptable for you.
